*J Physiol.* 2026 February ; 604(4): 1592–1606. doi:10.1113/JP287358.

# Gap junctional and ephaptic coupling in cardiac electrical propagation: homocellular and heterocellular perspectives

**Xiaobo Wu**[1], **Laura Beth Payne**[1], **Robert G. Gourdie**[1,2,3]

[1]Center for Heart and Reparative Medicine Research, Fralin Biomedical Research Institute at Virginia Tech Carilion, Roanoke, Virginia, USA

[2]Virginia Tech Carilion School of Medicine, Roanoke, Virginia, USA

[3]Department of Biomedical Engineering and Mechanics, Virginia Polytechnic Institute and State University, Blacksburg, Virginia, USA

## Abstract

Electrical communication in the heart is crucial for maintaining normal cardiac function. Traditionally, gap junctional coupling between cardiomyocytes has been accepted as the primary mechanism governing electrical propagation in the heart. However, numerous studies have demonstrated that gap junctions are also present between different cell types in heterocellular structures and disruption of such gap junctional coupling can be associated with cardiac dysfunction. In addition to gap junctional coupling, ephaptic coupling has been proposed as another mechanism for electrical communication between cardiomyocytes. Reducing ephaptic coupling has been shown to have negative impacts on cardiac conduction. While the existence of ephaptic coupling between different types of cardiac cell is under investigation, a recent study suggests that ephaptic coupling at heterocellular contacts between cardiomyocytes and fibroblasts may provide a proarrhythmic substrate in cardiac disease. In this review, we examine the current literature on electrical communication in the heart, including gap junctional and ephaptic coupling in homocellular and heterocellular contexts. Further, we offer a perspective on gaps in knowledge and opportunities for further advancing our understanding of electrical coupling mechanisms in action potential propagation in the heart.

## Graphical Abstract

Intricate cellular electrical coupling networks in the heart. Various cell types couple the central cardiomyocyte through gap junctional contacts, with the exception of neurons. Whether

**Corresponding author** Robert G. Gourdie: Fralin Biomedical Research Institute, Rm 2008, 2 Riverside Circle, Roanoke, VA 24016, USA. gourdier@vtc.vt.edu.

Author contributions

X.W. drafted the manuscript. L.B.P. contributed to the writing process by reviewing and editing the manuscript, providing critical feedback. R.G.G. supervised the project, provided critical revisions and approved the final version of the manuscript. All authors approved the final version of the manuscript. All persons designated as authors qualify for authorship, and all those who qualify for authorship are listed.

Competing interests

None declared.

ephaptic coupling (EpC) occurs in homocellular or heterocellular contexts beyond cardiomyocyte–cardiomyocyte interactions remains unclear.

## Keywords

ephaptic coupling; fibroblast; gap junctional coupling; heterocellular; homocellular

## Introduction

Electrical impulse propagation in the heart is a fundamental process that ensures the synchronized contraction of the myocardium, which is essential for maintaining normal cardiac function. Traditionally, gap junctions (GJs) between cardiomyocytes have been recognized as the primary mechanism responsible for action potential propagation in myocardial tissues (Hoagland et al., 2019; Nielsen et al., 2023; Rohr, 2004). However, connexins, the subunit proteins that form GJs in vertebrates, have been found to localize between cardiomyocytes and non-cardiomyocytes including cardiac fibroblasts, macrophages and endothelial cells (Goshima 1969; Johnson and Camelliti, 2018; Rook et al., 1992), suggesting that GJs probably form between cells apposed in such heterocellular contacts.

Research on ephaptic coupling (EpC) over the last decade, an old concept that has gained renewed interest, is challenging the previously dominant view on the role of gap junctional coupling (GJC) in cardiac conduction (Carmeliet 2019; Lin et al., 2010; Mori et al., 2008; Salvage et al., 2020; Sperelakis, 2002). Recent findings suggest that EpC may not only occur between cardiomyocytes, but is also possible between cardiomyocytes and other types of cell such as fibroblasts (Wang et al., 2023). Given these emerging insights into the electrical interactions within both homocellular and heterocellular associations in the heart, it seems timely to review current research on our understanding of the diverse mechanisms that govern cardiac electrical interactions.

In this review, we provide an overview of recent studies investigating cardiac electrical interactions in both homocellular and heterocellular contexts. Our further goal is to assist in identifying gaps in knowledge and suggest future research directions for developing

new understanding of the basis of arrhythmias and therapeutic strategies for treating these pathologies of cardiac conduction.

## Gap junctional coupling in homocellular contexts

### Gap junctional coupling between cardiomyocytes.

GJs are gated, highly conductive intercellular channels that are composed of protein subunits called connexins, which assemble into hexameric structures known as connexons or hemichannels in vertebrates (Lampe and Laird, 2022; Leybaert et al., 2017; Pun et al., 2022). So far, up to 21 different types of connexins in humans have been discovered, with four primary isoforms found in the heart including Cx37 (GJA4 gene on chromosome 1 in human), Cx40 (GJA5 on chromosome 1), Cx45 (GJA7 on chromosome 17) and Cx43 (GJA1 on chromosome 6), with Cx43 being the most abundant isoform overall (Palatinus et al., 2012).

Each connexin type contributes to the formation of GJ channels with distinct electrical properties, influencing the regulation, conductance and permeability of these channels. For example, cardiac connexins form channels with different levels of voltage sensitivity (van Veen et al., 2001), with Cx43 showing a modest response to transjunctional voltage, which appears to be reduced by phosphorylation of the protein (Ek-Vitorin et al., 2023). In the adult atria, Cx40 and Cx43 are prevalent, facilitating fast conduction and synchronized contraction (Lin et al., 2010). In the adult specialized pacemaking and conduction system, the main isoforms of connexin present, Cx40 and Cx45, form GJs varying in location and conductance (Delorme et al., 1997; Dhein and Salameh, 2021; Jansen et al., 2010; Lo, 2000). Cx45 is prominently expressed in sinoatrial and atrioventricular nodal tissues (Coppen et al., 1998, 1999; Yamamoto et al., 2006) and its channels exhibit low conductance of 30–40 pS (Elenes et al., 2001; Martinez et al., 2002). Cx40 is also found in the tissues of the His-Purkinje system, forming channels with high conductance of 180 pS (Beblo et al., 1995; Delorme et al., 1995; Gourdie, Green et al., 1993; Gourdie, Severs et al., 1993; Kanter et al., 1993; Yin et al., 2022). It has been suggested that compartmentalized expression and conductance of different connexin channels provides sequential propagation of electrical conduction (Gourdie et al., 1993; Lo 2000). In the mammalian ventricle, Cx43 channels with conductance of 90–120 pS are dominant, with evidence of substates (Elenes et al., 2001; Moreno et al., 1994), forming extensive networks that support the rapid propagation of the electrical impulses required for effective ventricular contraction (Beyer et al., 1987; van Veen et al., 2001; Verheule and Kaese, 2013).

It is widely held that GJC is the major determinant of electrical conduction in the heart. When an action potential is generated in one cardiomyocyte, the high-conductivity properties of GJs are hypothesized to allow ions, and small signalling molecules less than 1 kDa to quickly pass into the downstream cardiomyocyte, triggering depolarization and initiation of action potentials (Hoagland et al., 2019). GJs are primarily located within the interplicate regions of mature intercalated discs, which are oriented perpendicular to the long axis of the cell, particularly at the disc periphery, facilitating longitudinal conduction of electrical impulses (Gourdie et al., 1991). This process has long been conceived as

being crucial to the synchronous contraction of the heart muscle, which is necessary for maintaining consistent and effective electrical propagation.

While the integrity and function of GJs are held to be significant for maintaining normal cardiac rhythm, alterations in their abundance, distribution or function can lead to disturbances in cardiac electrophysiology, contributing to the development of arrhythmias and other cardiac diseases (Asimaki and Saffitz, 2012; Guo and Yang, 2022; Jongsma and Wilders, 2000). In myocardial ischaemia and infarction, one of the earliest reported changes was the lateralization of GJs at the border zones adjacent to infarct scars, accompanied by a reduction in Cx43 abundance levels (Peters et al., 1997; Severs, 1994; Smith et al., 1991). This remodelling of GJ distribution and connexin expression is conceived as contributing to the generation of a proarrhythmic substrate.

Mutations in the gene encoding Cx43, GJA1, can reduce Cx43 phosphorylation and induce loss of function of GJs, leading to decreased GJ conductance. This, in turn, is associated with slowed conduction and increases the risk of cardiac arrhythmogenesis (Kalcheva et al., 2007; Paznekas et al., 2003; Van Norstrand et al., 2012). By contrast, enhancing GJ communication has been reported to significantly decrease arrhythmia susceptibility in the setting of myocardial infarction, independent of infarct size (Ng et al., 2016). Ventricular arrhythmia resulting from spatial discordance in action potential alternans in ischaemia-induced cardiac arrest has also been shown to be reduced by upregulating GJ contact levels (Laurita et al., 2024). Therefore, maintaining GJ integrity appears to be required for normal cardiac function, and developing anti-arrhythmic strategies to specifically target GJs could be a path to providing benefit for patients with heart diseases.

### GJC between fibroblasts.

Fibroblasts in the heart are specialized cells that play a role in maintaining the structure and function of the heart's connective tissue (Dzhumagulov, 1989). They are responsible for producing and organizing the extracellular matrix (ECM), which provides structural support to cardiomyocytes. In addition to their function in maintaining the ECM, cardiac fibroblasts are also involved in repair processes following heart injury, such as myocardial infarction (Humeres and Frangogiannis, 2019; Wang et al., 2024).

It has been shown that GJs can form between fibroblasts with the type of connexin varying by location (Simon-Chica et al., 2023). Cx43 and Cx45 are expressed in fibroblasts in the ventricle (Gaudesius et al., 2003; Zhang et al., 2010), whilst Cx40 and Cx45 are mainly found in fibroblasts in the sinoatrial node (Camelliti et al., 2004; De Maziere et al., 1992). Many studies of fibroblast–fibroblast electrical interactions have been undertaken in cell culture, with demonstrations of GJs between fibroblasts by electron microscopy and measurements of channel conductance levels at ~22 pS (Doble and Kardami, 1995; Kohl and Gourdie, 2014; Rook et al., 1989), significantly lower than the 90–120 pS conductance found for GJs formed by Cx43 alone (Elenes et al., 2001).

GJs between fibroblasts have also been investigated in pathological conditions, including myocardial infarction (Camelliti et al., 2004). Kohl et al. (1994) demonstrated that Cx45-expressing fibroblasts acutely show a gradual increase following infarction, followed by

increases in Cx43-expressing fibroblasts in the infarcted region, suggesting a potential role for coupled fibroblasts in tissue remodelling following myocardial infarction. Notably, in a different study of myocardial infarction focused largely on use of primary cultures of cardiac cells, Zhang et al. (2010) observed up-regulated Cx43 in fibroblasts and down-regulated Cx43 in cardiomyocytes isolated from infarcted hearts. Intercellular communication between fibroblasts isolated from infarcted and non-infarcted regions, assessed by Lucifer yellow dye, was increased compared to fibroblasts isolated from sham-operated hearts. The findings from Zhang and co-workers suggest that fibroblasts up-regulate Cx43 to maintain interactions when electrical communication between cardiomyocytes is compromised.

Kohl and Gourdie (2014) posed the intriguing question, when they reviewed studies on fibroblast–myocyte electrotonic coupling: does it occur in native cardiac tissue? One might propose related questions on GJC between fibroblasts: does GJC between fibroblasts occur in native cardiac tissue? Do such homocellular non-myocyte interactions in the heart undergo modifications in pathological conditions such as ischaemic heart disease? With the development of new visualization techniques that improve resolution of the rare and small GJs that form between fibroblasts in native tissue, we may be able to better address such questions, improving our understanding of GJ structure and function between cardiac fibroblasts.

## GJC in heterocellular contexts

### GJC between cardiomyocytes and fibroblasts.

While GJs are well-studied in homocellular contexts, primarily between cardiomyocytes, numerous studies have demonstrated the presence of Cx43 and Cx45 in heterocellular interactions, particularly between cardiomyocytes and fibroblasts (Camelliti et al., 2005; Chilton et al., 2007; Gaudesius et al., 2003; Kohl et al., 1994; Mahoney et al., 2016; Rook et al., 1989). Further investigations have determined that GJs formed between cardiomyocytes and fibroblasts show modest conductance values, somewhat larger than those measured between fibroblasts, but smaller than those found between cardiomyocytes (Rook et al., 1989, 1992), which may explain decreased conduction rates observed when fibroblasts are co-cultured with cardiomyocytes (Nguyen et al., 2014; Rohr, 2009).

In addition to the well-established phenomenon of gap junctional communication between cardiomyocytes and fibroblasts, Gaudesius et al. (2003) determined that the presence of GJs formed by Cx43 or Cx45 supported synchronization of spontaneous activity in a strand of cardiomyocytes interconnected by fibroblasts, which may contribute to the generation of arrhythmias in fibrotic hearts due to large local conduction delays. Overall, these findings suggest that fibroblasts, which are traditionally considered structural and non-excitable cells, probably play a functional role in the heart's electrical network through their GJ connections with cardiomyocytes.

While functional GJs between cardiomyocytes and fibroblasts are well-established *in vitro*, it has been a matter of debate as to whether these exist *in vivo* probably due to the inability to reliably resolve these structures (Kohl and Gourdie, 2014). Optogenetic studies have provided electrophysiological evidence of heterocellular electrotonic coupling

in native myocardium, including myocyte–non-myocyte interactions, identifying tunnelling nanotubes as a possible substrate for such electrical cell interactions (Quinn et al., 2016). In a disease context, Fleischmann and colleagues demonstrated that when Cx43 was expressed exogenously in (myo)fibroblasts and CD45+ cells within myocardial infarction scars using lentiviruses in a mouse model, prominent and long-lasting arrhythmia protection was provided *in vivo* (Roell et al., 2018). Optical mapping of Cx43–virus-containing hearts revealed enhanced conduction velocity within the scar. The authors concluded that enhancement of Cx43-mediated electrical coupling between myocytes and scar-resident (myo)fibroblasts may account for these beneficial changes in diseased hearts.

Wang et al. (2023) recently provided further evidence that GJs can form between cardiomyocytes and fibroblasts *in vivo*. In their study, the authors genetically engineered a mouse that expressed a light-gated cation non-selective channel (Channelrhodopsin-2, ChR2) that could only be activated in fibroblasts (Wang et al., 2023). After myocardial infarction was induced, optical stimulation of the infarcted region elicited excitation and also induced arrhythmias. By contrast, optical stimulation of the uninjured cardiac regions failed to cause excitation spread. Overall, these findings suggest that electrical interactions can form between cardiomyocytes and fibroblasts *in vivo* and support the concept that remodelling of such contacts, particularly in pathological conditions, may be associated with cardiac arrhythmias (Nguyen et al., 2012, 2014).

### GJC between cardiomyocytes and macrophages.

Macrophages are the most abundant leukocytes in the heart (Pinto et al., 2016; Skelly et al., 2018), playing significant roles both in cardiovascular homeostasis and in pathophysiology (Gomez et al., 2018; Gourdie, 2022). Recent studies have demonstrated that 'resident cardiac macrophages' show functional presence of ionic channels, including voltage-gated potassium channels, as well as expression of Cx43, which can form GJs with cardiomyocytes (Bogert et al., 2024; Cao et al., 2024; Hulsmans et al., 2017; Simon-Chica et al., 2022). Computer simulations assessing the effects of macrophages electrotonically coupled to cardiomyocytes showed that these non-myocytes can depolarize resting cells, and shorten early and prolong late action potential duration, with effects depending on coupling strength and individual macrophage electrophysiological properties (Simon-Chica et al., 2022). The resting membrane potential and Kir2.1 level were found to be particularly important parameters in computational models.

Hulsmans et al. (2017) determined that macrophages isolated from atrioventricular (AV) conduction tissues were coupled with cardiomyocytes, and in turn, could affect their electrophysiological properties. Using transgenic mice, researchers found that ChR2-expressing macrophages can be optically manipulated to alter AV conduction. This approach allowed for precise control of macrophage electrical activity through light stimulation, demonstrating possible contributions to regulation of heart rhythm. Additionally, conditional deletion of Cx43 in cardiac macrophages disrupted conduction through the AV junction. Congenital ablation of macrophages also caused disturbances in AV conduction, highlighting possible assignments for these immune cells in maintaining normal electrical signalling in cardiac tissues.

Sugita et al. (2021) demonstrated that cardiac macrophages play a significant role in maintaining heart conduction during right ventricular stress. Using a mouse model of right ventricular pressure overload, they found that macrophages facilitate myocardial intercellular communication through GJs. Specifically, these immune cells produce amphiregulin (AREG), which regulates Cx43 phosphorylation and translocation in cardiomyocytes. Deletion of AREG in macrophages led to GJ disorganization and lethal arrhythmias during acute cardiac stress, highlighting the function of this macrophage-derived signal in preserving cardiac impulse conduction. Together, these observations suggest that heterocellular interactions between macrophages and cardiomyocytes could play a role in maintaining normal cardiac conduction, as well as in preserving electrical conduction in the context of disease.

### GJC between cardiomyocytes and endothelial cells.

The formation of GJs between cardiomyocytes and endothelial cells was demonstrated by Narmoneva et al. (2004). Their study found that when cardiomyocytes and endothelial cells were co-cultured, Cx43 expression between cardiomyocytes and endothelial cells significantly increased compared to cardiomyocytes cultured alone, alongside decreases in cardiomyocyte apoptosis and necrosis. In addition, it was observed that endothelial cells promoted synchronized contraction of cardiomyocytes, suggesting the formation of GJs between these two cell types. However, it remains unclear if gap junctional coupling occurs between these two types of cells in native tissue. As pointed out by others, the functional relevance of this type of heterocellular gap junctional contact *in vivo* is an interesting topic that would benefit from further study (Colliva et al., 2020).

## Ephaptic coupling in homocellular contexts

### Ephaptic coupling between cardiomyocytes.

Whilst GJC is widely considered to be the primary mechanism for electrical impulse propagation between excitable cells, numerous studies have suggested the importance of EpC in cardiac electrophysiology (George et al., 2016; Gourdie et al., 2021; Kucera et al., 2002; Lin and Keener, 2013; Mori et al., 2008; Veeraraghavan et al., 2018). EpC is a unique form of electrical communication that was first championed by Sperelakis and Mann (1977). Whilst there is ongoing debate, it is becoming accepted that EpC is a significant mechanism for electrical coupling in the heart, as well as the brain (Han et al., 2018; Lin et al., 2022; Martinez-Banaclocha 2018; Rebollo et al., 2021; Veeraraghavan et al., 2014). In contrast to GJ channels that directly couple two cells, EpC occurs at sites where two excitable cells are within nanometre-scale proximity such that electrical signals can be transmitted through specialized, narrow domains of extracellular space, providing the opportunity for local electric fields to influence the membrane potential of adjacent cells.

EpC between cardiomyocytes has been extensively investigated in computational models and whole-heart settings (Adams et al., 2023; Jaeger et al., 2019; Kucera et al., 2002; Lin et al., 2010; Morris et al., 2023; Otani et al., 2023; Veeraraghavan et al., 2018). Kucera et al. (2002) established a computational model investigating the involvement of EpC in cardiac conduction and concluded that this mechanism biphasically affects cardiac conduction,

depending on the cleft distance between two cells, as well as the level of GJ conductance. Rhett and Gourdie (2012) described a probable location for such an EpC-mediating cleft in ventricular myocardium known as the perinexus. The perinexus is a dynamic 20–30-nm-wide nanodomain of extracellular space within intercalated discs adjacent to GJs, expressing high densities of voltage-gated sodium channels (VGSCs), composed mainly of Nav1.5, and associated beta (e.g. $\beta 1/\beta 1b$) subunits (Fig. 1) (Kobirumaki-Shimozawa et al., 2020; Nielsen et al., 2023; Veeraraghavan et al., 2018; Williams et al., 2024).

In a more recent modelling study Ivanovic and Kucera (2021) showed that localization of sodium ($Na^+$) channel clusters in the perinexus is crucial for EpC, confirming that co-location with the GJ significantly modulates impulse transmission between cardiomyocytes. Varying the width of the perinexus greatly affected the interplay of $Na^+$ currents, extracellular potentials and patterns of current flow within the perinexal cleft. Moreover, during excitation it was found that the $Na^+$ current in the intercalated disc membrane of the pre-junctional cell switched from inward to outward, contributing ions to the activating channels on the post-junctional intercalated disc membrane.

In collaboration with Gourdie's group, Poelzing and co-workers have provided experimental confirmation that widening the perinexus significantly impacts cardiac conduction (George et al., 2015; Veeraraghavan et al., 2015, 2018). Further supported by work from others, including the aforementioned computational studies, the perinexus has emerged a probable pivotal site for EpC between cardiomyocytes (Adams et al., 2023).

Current studies indicate that EpC depends highly on two key conditions, both of which are met by the perinexus: (1) a narrow extracellular cleft between two closely apposed cells – appositions typically confined to intercalated discs in heart muscle; and (2) a high density of VGSCs. The mechanism of EpC is illustrated in Fig. 2. According to the EpC model, when two apposing cells are at rest, and in the absence of sodium channel activation, sodium ions diffuse through the tortuous extracellular space of the disc into perinexal domains at GJ edges (Ivanovic and Kucera, 2022; Moise et al., 2021; Salvage et al., 2020). Upon activation of sodium channels in the upstream cell (Cell 1), extracellular sodium ions flow rapidly into Cell 1, leading to ion depletion in the perinexus and a concomitant decrease in cleft potential ($V_p$). This cleft hyperpolarization induces sufficient post-junctional membrane depolarization in Cell 2, driving the post-junctional transmembrane potential closer to the sodium reversal potential. As a result, the reduced driving force for sodium current ($I_{Na}$) diminishes $I_{Na}$, ultimately slowing conduction – a phenomenon known as self-attenuation (George et al., 2016; Ivanovic and Kucera, 2021; Kucera et al., 2002; Moise et al., 2021). Adding to this concept, recent mathematical models have explored the effect of rate of sodium ion refilling of the perinexal cleft from the external extracellular space, demonstrating that regulation of this ion flux within the intercalated disc is key to maintenance of EpC (Morris et al., 2023).

According to the computational model of Kucera et al. (2002), when the perinexus is mildly widened from 20 nm (normal) to 50 nm, and gap junctional coupling is at the normal level representing 3–5%, the increased amount of sodium ions due to perinexal expansion can lead to an increase in sodium current in both cells, as well as the decrease

in cleft potential $V_p$, which enhances cardiac conduction. By contrast, if the perinexus is greatly widened beyond 50 nm, the sodium ions in the perinexus cannot be depleted after activation of sodium channels in Cell 1 due to the excess of ions available, and this reduced self-attenuation only causes slight decreases in cleft potential, which in turn fails to increase the post-junctional membrane potential of Cell 2 to the threshold for sodium channel activation, thereby leading to slowing of cardiac conduction.

Detailed descriptions of EpC mechanisms and their implication for normal and disrupted cardiac conduction can be found in publications from multiple labs (Kucera et al., 2002; Lin et al., 2010; Moise et al., 2021; Otani et al., 2023). It should be noted that a majority of studies have focused on VGSCs when investigating EpC. However, voltage-gated potassium channels are also found in the intercalated disc and have been suggested to be involved in EpC (Ly and Weinberg, 2022; Poelzing et al., 2021; Sperelakis and McConnell, 2002; Veeraraghavan et al., 2016).

EpC between cardiomyocytes in multiple pathological conditions has been investigated. The first study surveying the perinexus in diseased human hearts demonstrated that perinexal nanodomains were widened in patients with persistent atrial fibrillation relative to healthy people (Raisch et al., 2018). A mechanism for how these nanodomains become disrupted in atrial arrhythmias was suggested by studies in mice showing that inhibition of inflammation-induced vascular leak prevented perinexal widening and loss of sodium channels (Mezache et al., 2023).

In the setting of cardiac ischaemic injury, widening of the perinexus attenuated conduction slowing, suggesting the involvement of EpC in adaptive responses to ischaemic stress (Hoeker et al., 2020). Others have also pointed to a role of EpC and the complex geometry of the border zone in arrhythmia termination during acute myocardial ischaemia (Wei and Tolkacheva, 2022). Widening of the perinexus has also been reported to significantly prolong cardiac action potential duration (APD) and increase age-related arrhythmogenesis in a guinea pig model of sodium channel gain-of-function, known as long QT syndrome type 3 (LQT3) (Greer-Short et al., 2017; Nowak et al., 2020, 2021; Wu et al., 2021, 2025), and particularly with hypernatremic stress, which further exacerbates the LQT3 phenotype (Wu et al., 2021). Overall, these studies highlight the important role of the perinexus in mediating a pathological phenotype in a manner consistent with EpC.

### EpC between fibroblasts.

EpC between fibroblasts in the heart has not yet been explored, and thus it is unknown if EpC can occur in this setting. As described earlier, the incidence of cardiac EpC as currently envisaged requires both a high density of sodium channels and a narrow cleft located between two cells. Some key ionic channel proteins related to EpC have been found in fibroblasts, including Nav1.5 and Kir2, which should contribute to electrogenic activity (Chatelier et al., 2012; Li et al., 2009). In addition, as already outlined, fibroblasts express connexins and form GJs (Chatelier et al., 2012; Gaudesius et al., 2003), potentially underpinning inter-fibroblast electrical transfer (Chacar et al., 2017; Rook et al., 1989). Therefore, conditions necessary for EpC appear to be met between fibroblasts.

This being said, the regions between fibroblasts where GJs may be present have not been closely investigated and it remains unknown if a specialized nanodomain analogous to the perinexus occurs at such contacts. Future study might focus on characterizing the ultrastructure of shared intercellular structures occurring between fibroblasts. It also has to be considered that as yet unknown EpC mechanisms may occur between fibroblasts, as well as in other homocellular and heterocellular contexts. For example, a unique EpC mechanism involving an unconventional role for connexin hemichannels and calcium currents has been identified in the zebrafish retina (Klaassen et al., 2011) – thus future hypotheses should be open to the possibility of more than one type of EpC mechanism occurring in the heart.

## EpC in heterocellular contexts

### EpC between cardiomyocytes and fibroblasts.

The possibility of EpC between cardiomyocytes and fibroblasts was proposed by Kohl and Gourdie (2014). However, Wang et al. (2023) experimentally investigated such an EpC mechanism in a mouse model of myocardial infarction. In their study, Wang and colleagues genetically engineered a mouse expressing ChR2 exclusively in cardiac fibroblasts. The authors found that optical stimulation of scar tissue elicited organ-wide cardiac excitation and induced arrhythmias, suggesting fibroblasts electrically communicate with cardiomyocytes. Interestingly, after knocking out all known cardiac GJ-forming proteins in cultures of neonatal myocytes containing the optogenetic channel, including Cx40, Cx43, Cx45 and pannexin 1, optical stimulation continued to lead to widespread cardiac excitation, suggesting that electrical coupling between fibroblasts and cardiomyocytes may be mediated by EpC. A computational model was used to further investigate electrical interactions between cardiomyocytes and fibroblasts, which indicated that GJC and EpC may excite myocytes coupled to fibroblasts in a synergistic, yet functionally redundant manner.

The study by Wang et al. (2023) provides new insights into the mechanisms of GJC and EpC between cardiomyocytes and fibroblasts *in vivo*. However, several questions remain including: (1) What are the specific nanostructures responsible for this form of heterocellular EpC? (2) Can scar fibroblasts form shared VGSC complexes with cardiomyocytes? (3) Do narrow perinexus-like clefts of extracellular space form between fibroblasts and cardiomyocytes? Also, whilst the current computational model provides an initial investigative framework, its architecture will probably need to undergo iterative refinement as mechanistic insights into fibroblast–myocyte EpC dynamics evolve. Key parameters – notably sodium channel conductivity in fibroblast populations – warrant targeted validation through both sensitivity analyses and empirical electrophysiological studies to further strengthen predictive accuracy. It should further be considered that the EpC processes that occur between cardiomyocytes and fibroblasts may differ in structure and function fundamentally from those between cardiomyocytes. Addressing these questions is crucial, as fibroblasts have been shown to contribute to cardiac arrhythmogenesis in pathological conditions, particularly in fibroblast–myocyte interaction-rich infarct border zones (Hall et al., 2021; Nguyen et al., 2012; Rubart et al., 2018; Verheule and Schotten, 2021). Elucidating the mechanisms of electrical coupling at such pro-arrhythmic substrates could lead to the development of novel anti-arrhythmic therapies.

### EpC between cardiomyocytes and other types of cells.

Heterocellular EpC between cardiomyocytes and cell types other than fibroblasts including macrophages, and endothelial or cardiac nerve cells is yet to be explored (Qu et al., 2024; Simon-Chica et al., 2023). Similar to the unanswered questions regarding EpC between fibroblasts, there is uncertainty about the existence of EpC-mediating structures between cardiomyocytes and other such cell types. Although both macrophages and endothelial cells can express VGSC subunits and various potassium channels, including Kir2.1 (Gourdie 2022; Hulsmans et al., 2017; Jackson 2022), which have been implicated in electrical signalling processes, the specific presence and role of such channels in facilitating EpC between cardiomyocytes and these other cell types remains unclear.

In a related vein, the interaction between cardiomyocytes and nerves remains mysterious and intriguing. Unlike skeletal muscle, where peripheral nerve cells form well-defined neuromuscular junctions, no such specialized junctions have been clearly identified in cardiomyocytes (Hastings and Valdez, 2024). The precise structural basis of neuron–myocyte interactions remains unknown. However, a recent computational study of the vestibular system predicts the concurrence of synaptic and ephaptic coupling in the hair cell–calyx system (Govindaraju et al., 2023), where sodium channels have been observed at synapses (Leao et al., 2005; Wang et al., 2017). This raises the possibility that ephaptic coupling may also occur between neurons and cardiomyocytes. Given the abundance of cardiac nerves and their critical role in arrhythmia formation (Ripplinger et al., 2016; Zhu et al., 2022), further investigation into their potential EpC interactions is warranted.

## Future directions and conclusions

Our review highlights the complexity of cardiac electrical coupling, encompassing both homocellular and heterocellular interactions via GJC and EpC. A companion review in this issue of the *Journal of Physiology* by Simon-Chica and colleagues covers a related topic on modelling of cardiomyocyte–non-myocyte electrical interactions (Simon-Chica et al., 2025). Recent research has expanded our understanding beyond cardiomyocyte-centric models, revealing significant roles for fibroblasts, macrophages and endothelial cells in cardiac conduction. A summary of known and putative electrical interactions between cardiac cells is illustrated in Fig. 3. Key questions remain, including the relationship between GJC and EpC. Are GJC and EpC independent mechanisms, which nonetheless interact redundantly to support cardiac conduction, or does the GJ and perinexus specify a single integrated mechanism for this purpose? Importantly, a detailed examination of ion dynamics in the perinexus within its 3D structure will enhance our understanding of the diffusion process, such as how sodium ions access the perinexus during self-attenuation, thereby elucidating the mechanisms of ephaptic coupling at the nanoscale. There is also consideration of the impact of heterocellular electrical interactions on non-myocyte function. Notably, the influence of cardiomyocyte–fibroblast electrical coupling on fibroblast differentiation and collagen secretion remains underexplored. These interactions may alter fibroblast behaviour beyond merely affecting conduction velocity, potentially impacting pathological remodelling of cardiac tissues and thereby have indirect effects on arrhythmia susceptibility. The role

of electrical coupling in specialized cardiac tissues (e.g. His-Purkinje system) and during development are further areas that warrant investigation.

This being said, current techniques impose significant limitations on our ability to fully understand the mechanisms of electrical coupling, particularly EpC, at the cellular level. One major challenge is determining whether gap junctional and ephaptic couplings function as truly independent mechanisms. Existing techniques cannot ensure the complete knockout of all connexin types involved in gap junctional coupling within cardiac tissue. Although CRISPR-Cas9 gene editing is a promising tool, achieving the knockout of all connexins in cardiomyocytes or other relevant cell types remains challenging. Moreover, there is currently no method available to directly track sodium ion diffusion into the perinexus. A potential approach could involve a modified fluorescence resonance energy transfer (FRET) system, where labelled sodium ions act as donors and perinexal sodium channels serve as acceptors, allowing visualization of sodium ion movement near these channels. Further advancements in imaging, electrophysiology and molecular tools are essential for overcoming these limitations and gaining deeper insights into coupling mechanisms.

The knowledge gaps pointed to in this review may be part of the explanation as to why there has been limited progress in developing new arrhythmia therapies over the past three decades (Saljic et al., 2023). Current anti-arrhythmic agents have limited efficacy and potential pro-arrhythmic side effects (Cardiac Arrhythmia Suppression Trial, 1992; Echt et al., 1991), underscoring the need for new, safe and effective therapeutic strategies. Advancing our understanding of cardiac conduction mechanisms, particularly those associated with heterocellular interactions in the heart, could lead to more effective and targeted anti-arrhythmic treatments. This research direction not only promises to deepen our knowledge of cardiac electrophysiology, but also holds potential for significant clinical impact in managing cardiac arrhythmias.

## Supplementary Material

Refer to Web version on PubMed Central for supplementary material.

## Acknowledgements

The authors have no acknowledgements to declare.

### Funding

This work was supported by the Lyerly Postdoctoral Excellence Award (to X.W.) from Fralin Biomedical Research Institute, the National Heart, Lung, and Blood Institute Grants 1R35HL161237–01 (to R.G.G.), and gifts from the Fralin family (Heywood Fralin Professorship) and Redgates Foundation to R.G.G.

## Biography

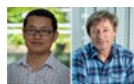

**Xiaobo Wu** earned his PhD in Translational Biology, Medicine and Health at Fralin Biomedical Research Institute (FBRI) at Virginia Tech. He is currently a postdoctoral

associate in Dr Gourdie's lab at FBRI. His research focuses on electrical coupling in the heart. **Robert Gourdie**, the Heywood Fralin Professor at Virginia Tech, leads the Vascular and Heart Research Center at the Fralin Biomedical Research Institute. He earned his bachelor's degree in Cell and Molecular Biology from the University of Auckland and PhD from the University of Canterbury in New Zealand. Following training as a British Heart Foundation Fellow at University College London, Gourdie continued to focus on connexin-based signalling in development, health and disease. His research encompasses cardiac ephaptic conduction and exosome-based therapies for myocardial infarction and radiation injury, bridging fundamental cellular communication studies with clinical and therapeutic applications.

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

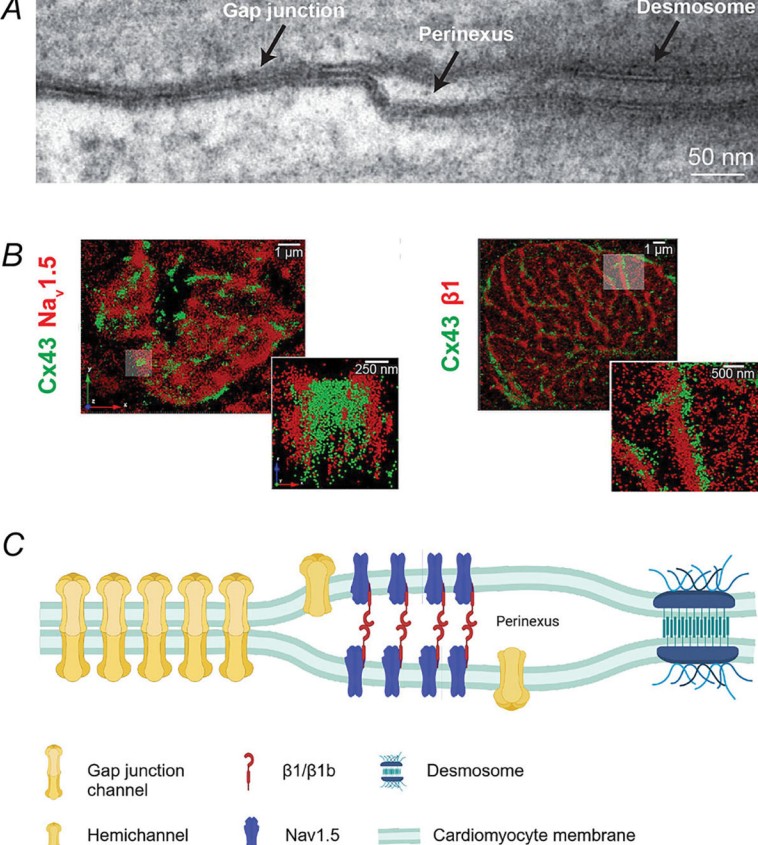

**Figure 1. The perinexus is localized adjacent to gap junctions in the intercalated discs between two cardiomyocytes**

*A*, a perinexus is shown located between a GJ and a desmosome in an electron micrograph of a region of the intercalated disc (ID) in the ventricle. The perinexal nanodomain is a narrow region of extracellular space shared between ventricular cardiomyocytes at the ID. *B*, Nav1.5 and $\beta$1 subunits in perinexal nanodomains next to a Cx43 GJ, as revealed in an *en face* ID, imaged by stochastic optical reconstruction (super-resolution) microscopy (STORM). *C*, a model of the perinexal region in the ID.

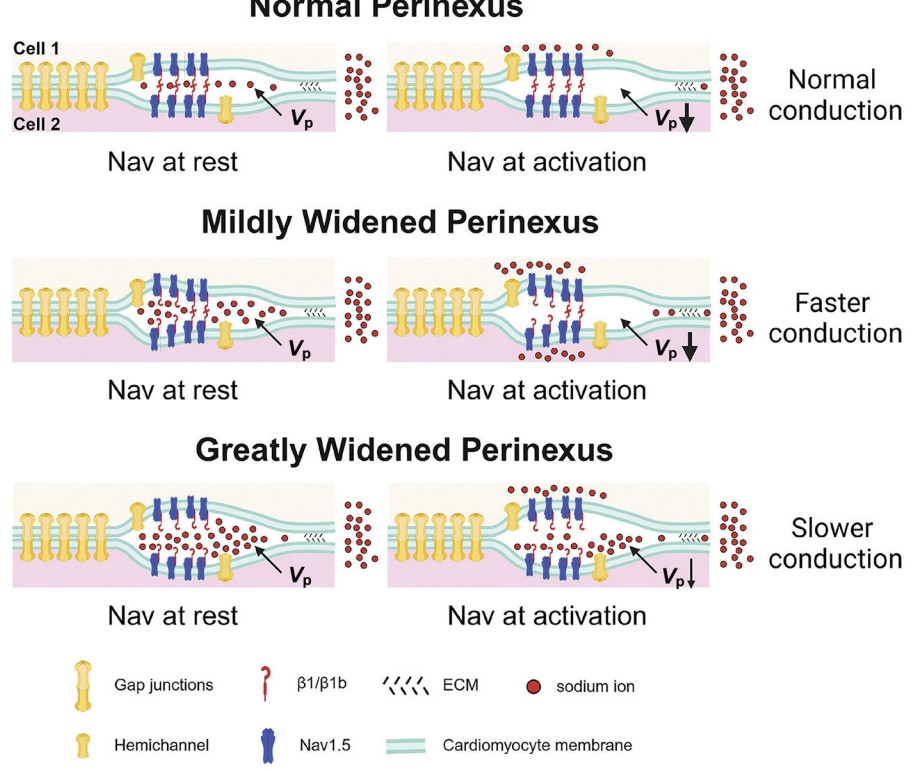

**Figure 2. Modes of EpC mediated by varying perinexal width**

The literature provides support that EpC occurs at the perinexus – a nanodomain of extracellular space within an ID shared between ventricular cardiomyocytes. The diagram interprets the complex mechanisms of EpC that may operate under normal, mild and high levels of widening of perinexal nanodomains. Briefly, according to the driving force of sodium current (Ivanovic and Kucera 2021): $I_{Na–cleft} = G_{Na} * (V_i – V_p – E_{Na})$, where $I_{Na\text{-}cleft}$ is the perinexal sodium current flowing into Cell 1 or Cell 2 through the perinexus, $G_{Na}$ is the sodium channel conductance, $V_i$ is the intracellular potential, $V_p$ is the cleft potential and $E_{Na}$ is the reversal potential for sodium ions, when sodium channels in Cell 1 are activated, sodium ions in the perinexus rapidly flow into Cell 1, leading to depletion of sodium ions in the normal perinexus. As a result, no available sodium ions in the perinexus enter Cell 2, even if the post-junctional membrane potential of Cell 2 reaches the sodium channel activation threshold due to greatly reduced $V_p$. When the perinexus is mildly widened, sodium ions remain available after activation of sodium channels in Cell 1 and they can rapidly enter Cell 2, leading to increased conduction compared to the normal perinexus. By contrast, when the perinexus is greatly widened, the influx of sodium ions into Cell 2 becomes slow as the post-junctional membrane potential fails to reach the sodium channel activation threshold due to a modestly reduced $V_p$, leading to slow conduction.

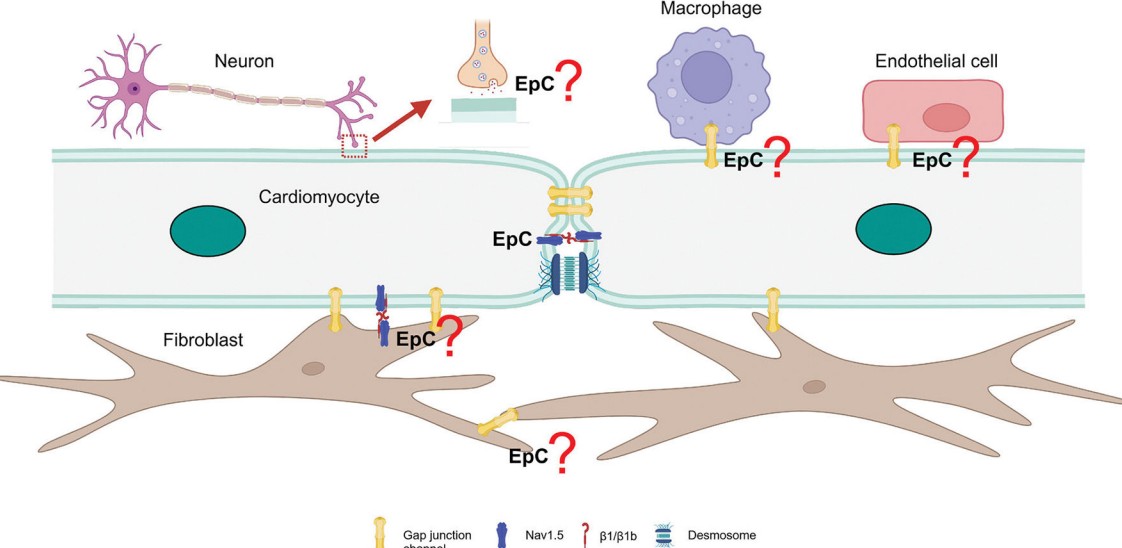

**Figure 3. Knowns and unknowns in the cellular electrical coupling networks found in the heart**
An endothelial and nerve cell and fibroblast and macrophage couple onto the
centred cardiomyocyte through gap junctional contacts. Growing evidence supports that
homocellular EpC occurs between cardiomyocytes at the perinexus, but it remains unclear
whether this mechanism operates at other types of heterocellular interaction. The existence
and/or mechanism of electrical interaction between neurons and cardiomyocytes remains an
intriguing unknown.

