## [Peer Review File · The Journal of physiology]

Gap Junctional and Ephaptic Coupling in Cardiac Electrical Propagation: Homocellular and Heterocellular Perspectives

Xiaobo Wu, Laura Beth Payne, and Robert G. Gourdie

DOI: 10.1113/JP287358

Corresponding author(s): Robert Gourdie (gourdier@vtc.vt.edu)

The following individual(s) involved in review of this submission have agreed to reveal their identity: Franziska Schneider-Warme (Referee #2)

Review Timeline:

Submission Date:	16-Dec-2024
Editorial Decision:	27-Jan-2025
Revision Received:	28-Mar-2025
Accepted:	28-Apr-2025

Senior Editor: Bjorn Knollmann

Reviewing Editor: T Alexander Quinn

Transaction Report:

Dear Dr Gourdie,

Re: JP-TR-2024-287358 "Homocellular and heterocellular electrical interactions in the heart" by Xiaobo Wu, Laura Beth Payne, and Robert G Gourdie

Thank you for submitting your manuscript to The Journal of Physiology. It has been assessed by a Reviewing Editor and by 2 expert referees and we are pleased to tell you that it is acceptable for publication following satisfactory revision.

Please address all the points raised and incorporate all requested revisions or explain in your Response to Referees why a change has not been made. We hope you will find the comments helpful and that you will be able to return your revised manuscript within two months. If you require longer than this, please contact journal staff: jp@physoc.org.

ABSTRACT FIGURES: Authors may use The Journal's premium BioRender account to create/redraw their Abstract Figures (and any other suitable schematic figure). Information on how to access this account is here: <https://physoc.onlinelibrary.wiley.com/journal/14697793/biorender-access>.

REVISION CHECKLIST: Upload a full Response to Referees file. To create your 'Response to Referees' copy all the reports, including any comments from the Senior and Reviewing Editors, into a Microsoft Word, or similar, file and respond to each point, using font or background colour to distinguish comments and responses and upload as the required file type.

We look forward to receiving your revised submission.

Yours sincerely,

Bjorn Knollmann
Senior Editor

EDITOR COMMENTS

Reviewing Editor:

Your paper has been reviewed by two experts in the field, who both felt it is a well-written, comprehensive review, that will make a valuable addition to the literature. They did, however, have suggestions to improve the manuscript, which should be addressed in a revised version. Please update the manuscript accordingly, including a point-by-point response to the reviewers' concerns.

Senior Editor:

I concur with the reviewing editor.

REFeree COMMENTS

Referee #1:

This is a timely review from Wu and colleagues on the role of electrical interactions in the heart, considering both homocellular and heterocellular forms of coupling. This is a well-written and extensive review, capturing the latest in both experimental and computational studies.

I have mostly minor issues for the authors for additional consideration, with point 1 most critical:

1. The description of 'self-attenuation' on lines 270-271 is not quite correct. Self-attenuation does not specifically refer to the process of sodium ion depletion in the cleft. Rather self-attenuation refers to the process in which cleft hyperpolarization results in sufficient postjunctional membrane depolarization such that the postjunctional transmembrane potential approaches the sodium reversal potential. This polarization results in a reduction in the INa driving force, which in turn reduces the INa current and drives subsequent conduction slowing.

These processes are related, as cleft sodium ion depletion will also reduce the sodium reversal potential. Note that ref 79 is the first to use the term 'self-attenuation' but does not explicitly model dynamic extracellular cleft sodium concentration (and thus cannot reproduce sodium ion depletion).

2. On lines 288-291, the authors note the potential role of potassium channels in ephaptic coupling. Two notable key studies (PMIDs 34757078 and 35364829), both relevant computational studies, consider how potassium current localization at the intercalated disc can mediate automaticity through ephaptic coupling.

3. On lines 364-365, the authors comment on the neuro-muscular junction and the potential for ephaptic coupling communication. I agree that this is an interesting and novel area to highlight. A notable recent computational study (PMID 36595693) of the vestibular system predicts the concurrence of synaptic and ephaptic coupling (referred to in this study as nonquantal transmission) in the hair cell-calyx system.

Referee #2:

The submitted review article by Drs Wu, Payne, and Gourdie provides a comprehensive review on homo- and heterocellular electrical coupling and the proposed underlying mechanisms, summarizing the relevant literature and the current state-of-knowledge.

Please find my detailed suggestions for improving the manuscript (minor revisions) below.

- The title of the manuscript states "electrical interactions", whereas to my understanding the provided manuscript focusses almost exclusively on mechanisms of gap-junction-mediated and ephaptic electrical coupling. I would therefore suggest to further focusing the title of the manuscript. (For example, aspects such as electrical insulation by fibroblasts or neuron-supporting cells are not discussed, also the functional relevance of electrical interactions for the individual cell types beyond their influence on cardiac impulse propagation is not discussed in detail.)
- The manuscript discusses in detail a recent study showing Cx-dependent and independent coupling between cardiac scar-resident fibroblasts and cardiomyocytes (Wang et al., reference 12). While the data presented in this study is indeed intriguing, the study has been very controversially discussed, for example because of light intensities used that are 100-1000-fold higher than those necessary to effectively stimulate Channelrhodopsin-2, but also because the computational modelling data builds on parameters (e.g. Na channel conductivity in fibroblasts), that are beyond values assumed to be realistic in the field. As already highlighted in the current manuscript, the presence of perinexus-like structures at the interfaces between fibroblasts and cardiomyocytes has not been shown, thus the mechanisms underlying the Wang study are unclear at present. I would therefore suggest to even more cautiously discuss data from this paper, highlighting limitations of the study and lack of mechanistic understanding.
- The in-depth discussion of the proposed mechanisms of ephaptic coupling is important. However, in the current form, I cannot follow the discussion on "Na ion depletion" in the perinexus. Has this been experimentally validated, e.g. by use of Na-concentration dependent fluorophores or Na-selective electrodes? Given the very high driving force for Na import, it is hard to imagine that the gradient can be depleted (only very few ions need to be transported for large transmembrane voltage changes to occur), even within diffusion-limited nanodomains. Would one not also expect a change in extracellular K concentration? Another interesting aspect in this regard is how the perinexus is organised in species with different heart rates (e.g. mouse vs human)?
- The manuscript would benefit from a short discussion on methodological advances that would facilitate future investigations into electrical coupling mechanisms. Which methods need further development or could be adapted already to fill open knowledge gaps? Some suggestions: Nanodomain functional imaging of diffusion or changes in ion concentrations, 3D structural methods at nano- and microscale to characterize cell-cell interaction sites, optogenetic actuators targeted to specific cellular compartments etc.
- In certain places, the manuscript would benefit from more clearly defined terminology and/or by the specification of quantitative measures.
 - o Line 25: Please specify "heterocellular structures"
 - o Line 44: You may want to reconsider the term "communication", I would suggest something like "electrical impulse propagation" .
 - o Line 47: "Electrical propagation" - please specify what is propagated, e.g. "action potentials" or "depolarization/repolarization cycles".
 - o Lines 82-84: Please provide quantitative values for low and high conductances
 - o Line 93: You may want to add a statement of intercalated disc orientation and directionality of conduction here. You later introduce lateralization of connexins, but this may not be clear without introducing the normal distribution of connexins in myocytes (e.g. ratio between short and long axes).
 - o Line 107: Can you please specify "reductions in CV"? Are these due to changes in connexin expression, localization, and/or conductivity of individual gap junctions?
 - o Lines 126/127: "reports of both structural GJs and conductance and electrical activity" - Please phrase more clearly.
 - o Lines 137: Please specify comparison: "showed increased GJC" compared to XXX.
 - o Lines 145: "techniques and approach" - Please correct grammar and please specify the techniques you are referring to.
 - o Lines 156/157: "Fibroblasts are expressed in cultured cardiomyocytes" - Please rephrase (co-expression model).
 - o Line 164: "Active participants" - term could be misleading. Do you refer to the presence of voltage-gated Na channels in fibroblasts? Otherwise I would use different terminology to highlight the role of fibroblasts for cardiac impulse propagation (not initiation).
 - o Lines 175: Would replace "scar" by "scar-resident".

- o Line 179: I would propose to properly introduce Channelrhodopsin-2 here (currently only introduced on line 200) - light-gated cation non-selective channel (Channelrhodopsin-2)
 - o Line 185: "is" - would suggest to replace by "can be".
 - o Line 221: "Cx43 expression significantly increased" - please specify in which cell type.
 - o Lines 223/224: "suggesting a role for GJs between these two cell types in spread of electrical activation" - description of evidence and thus conclusion is not very clear. Is there any direct evidence of (gap junctional) coupling between the two cell types, or could the interaction be indirectly (e.g. signalling based)?
 - o Lines 229/230: "primary mechanism for electrical interaction between excitable cells" - see comment on title; I think interaction is too general and then the statement might be debatable.
 - o Line 235: "sufficiently close proximity" - please specify.
 - o Line 279: "mildly widened" - please specify. In the schematic Figure, there are two charges more than for normal, but what this represents in terms of distance/area/volume or actual numbers of charges is not very clear.
 - o Line 312: Would the Na channel density in fibroblasts be high enough for effective ephaptic coupling?
 - o Line 330: Please use proper terminology when discussing findings on ChR2 (see comment above).
 - o Line 337: "these workers" - please rephrase
 - o Line 341/342: Please reconsider phrasing of statement as study is highly controversial (see comment above).
- The Figures could be improved to further support the presented concepts.
- o Figure 1B: Please double-check scale bars of overview images and insets. The inset on the left seems to be more than 2-fold increased compared to the overview image. Does the highlighted region correspond 1:1 to the zoom-in?
 - o Line 404: "is narrow" - Please correct grammar and specify dimension if possible.
 - o In the context of Figure 1 you may want to highlight the need for 3D nanoscopic investigations to better understand diffusion-limited domains, their access and volumes.
 - o Figure 2: The difference between the upper and middle row is difficult to grasp, and the faster conduction (middle row) is not shown directly. Providing a voltage value always requires a reference, you may want to add this to the legend.
 - o Figure 3: Please use more realistic representations of cardiac interstitial cells, as well as endothelial cells and axons (not necessarily terminating on cardiomyocytes, but having axonal contacts along their entire length). In intact myocardium, these cells all form relatively thin processes wrapping around cardiomyocytes (filling the "gaps"). E.g. see <https://pubmed.ncbi.nlm.nih.gov/37674983/> <https://www.biorxiv.org/content/10.1101/2023.11.30.569388v1>
- Finally, there are a numbers of typos to be corrected.
- o Line 91: please remove extra "and"
 - o Line 126: please remove extra comma
 - o Line 216: "play role"
 - o Line 219: "That formation" (incomplete sentence)
 - o Line 249: Please correct: "Kucera"
 - o Line 260: Extra space.
 - o Figure 1A: Please add space between number and unit.

REQUIRED ITEMS

- Please include an Abstract Figure file, as well as the Figure Legend text within the main article file. The Abstract Figure is a piece of artwork designed to give readers an immediate understanding of the Review Article and should summarise the main conclusions. If possible, the image should be easily 'readable' from left to right or top to bottom. It should show the physiological relevance of the Review so readers can assess the importance and content of the article. Abstract Figures should not merely recapitulate other figures in the Review. Please try to keep the diagram as simple as possible and without superfluous information that may distract from the main conclusion of the Review. Abstract Figures must be provided by authors no later than the revised manuscript stage and should be uploaded as a separate file during online submission labelled as File Type 'Abstract Figure'. Please ensure that you include the figure legend in the main article file. All Abstract Figures will be sent to a professional illustrator for redrawing and you may be asked to approve the redrawn figure before your paper is accepted.

- Your MS must include a complete "Additional information section" with the following 4 headings and content:

Competing Interests: A statement regarding competing interests. If there are no competing interests, a statement to this effect must be included. All authors should disclose any conflict of interest in accordance with journal policy.

Author contributions: Each author should take responsibility for a particular section of the study and have contributed to writing the paper. Acquisition of funding, administrative support or the collection of data alone does not justify authorship; these contributions to the study should be listed in the Acknowledgements. Additional information such as 'X and Y have contributed equally to this work' may be added as a footnote on the title page.

It must be stated that all authors approved the final version of the manuscript and that all persons designated as authors qualify for authorship, and all those who qualify for authorship are listed.

Funding: Authors must indicate all sources of funding, including grant numbers. If authors have not received funding, this must be stated.

It is the responsibility of authors funded by RCUK to adhere to their policy regarding funding sources and underlying research material. The policy requires funding information to be included within the acknowledgement section of a paper. Guidance on how to acknowledge funding information is provided by the Research Information Network. The policy also requires all research papers, if applicable, to include a statement on how any underlying research materials, such as data, samples or models, can be accessed. However, the policy does not require that the data must be made open. If there are considered to be good or compelling reasons to protect access to the data, for example commercial confidentiality or legitimate sensitivities around data derived from potentially identifiable human participants, these should be included in the statement.

Acknowledgements: Acknowledgements should be the minimum consistent with courtesy. The wording of acknowledgements of scientific assistance or advice must have been seen and approved by the persons concerned. This section should not include details of funding.

- The reference list must be in alphabetical order, rather than numbered, to comply with our Journal format.

- Please upload separate high quality figure files via the submission form.

- Author profile(s) must be uploaded via the submission form. Authors should submit a short biography (no more than 100 words for one author or 150 words in total for two authors) and a portrait photograph of the two leading authors on the paper. These should be uploaded and clearly labelled together in a Word document with the revised version of the manuscript. Any standard image format for the photograph is acceptable, but the resolution should be at least 300 DPI and preferably more. A group photograph of all authors is also acceptable, providing the biography for the whole group does not exceed 150 words.

END OF COMMENTS

Dear reviewers,

Thank you for the valuable feedback and constructive suggestions on our manuscript. We have carefully reviewed all your comments and have revised the manuscript accordingly. Our responses to each of your points are provided in detail below. We hope that our responses and the revisions address your concerns. We appreciate your time and effort in helping us improve the quality of this paper.

EDITOR COMMENTS

Reviewing Editor:

Your paper has been reviewed by two experts in the field, who both felt it is a well-written, comprehensive review, that will make a valuable addition to the literature. They did, however, have suggestions to improve the manuscript, which should be addressed in a revised version. Please update the manuscript accordingly, including a point-by-point response to the reviewers' concerns.

Senior Editor:

I concur with the reviewing editor.

REFEREE COMMENTS

Referee #1:

This is a timely review from Wu and colleagues on the role of electrical interactions in the heart, considering both homocellular and heterocellular forms of coupling. This is a well-written and extensive review, capturing the latest in both experimental and computational studies.

I have mostly minor issues for the authors for additional consideration, with point 1 most critical:

1. The description of 'self-attenuation' on lines 270-271 is not quite correct. Self-attenuation does not specifically refer to the process of sodium ion depletion in the cleft. Rather self-attenuation refers to the process in which cleft hyperpolarization results in sufficient postjunctional membrane depolarization such that the postjunctional

transmembrane potential approaches the sodium reversal potential. This polarization results in a reduction in the I_{Na} driving force, which in turn reduces the I_{Na} current and drives subsequent conduction slowing.

These processes are related, as cleft sodium ion depletion will also reduce the sodium reversal potential. Note that ref 79 is the first to use the term 'self-attenuation' but does not explicitly model dynamic extracellular cleft sodium concentration (and thus cannot reproduce sodium ion depletion).

Thank you for your valuable feedback. We have revised the definition of self-attenuation according to your suggestion. Additionally, we included a reference (PMID: 34533834), which models cleft sodium ion dynamics. This revised sentence is provided here for your convenience:

This cleft hyperpolarization induces sufficient post-junctional membrane depolarization in Cell 2, driving the post-junctional transmembrane potential closer to the sodium reversal potential. As a result, the reduced driving force for sodium current (I_{Na}) diminishes I_{Na} , ultimately slowing conduction—a phenomenon known as self-attenuation (Kucera et al. 2002, George et al. 2016, Ivanovic et al. 2021, Moise et al. 2021).

2. On lines 288-291, the authors note the potential role of potassium channels in ephaptic coupling. Two notable key studies (PMIDs 34757078 and 35364829), both relevant computational studies, consider how potassium current localization at the intercalated disc can mediate automaticity through ephaptic coupling.

Thank you for pointing out these relevant computational studies. Both studies have now been cited.

3. On lines 364-365, the authors comment on the neuro-muscular junction and the potential for ephaptic coupling communication. I agree that this is an interesting and novel area to highlight. A notable recent computational study (PMID 36595693) of the vestibular system predicts the concurrence of synaptic and ephaptic coupling (referred to in this study as nonquantal transmission) in the hair cell-calyx system.

Thank you for providing this important information. We have revised the relevant sentences to incorporate the recommended citation, along with additional references confirming the presence of sodium channels at synapses. The revised sentences are also provided below for your convenience. In addition, we also revised Figure 3 to label the uncertain existence of EpC between neurons and cardiomyocytes. We hope these revisions address your concerns.

In a related vein, the interaction between cardiomyocytes and nerves remains mysterious and intriguing. Unlike skeletal muscle, where peripheral nerve cells form well-defined neuromuscular junctions, no such specialized junctions have been clearly identified in cardiomyocytes (Hastings et al. 2024). The precise structural basis of neuron-myocyte interactions remains unknown. However, a recent computational study of the vestibular system predicts the concurrence of synaptic and ephaptic coupling in the hair cell-calyx system (Govindaraju et al. 2023), where sodium channels have been observed at synapses (Leao et al. 2005, Wang et al. 2017). This raises the possibility that ephaptic coupling may also occur between neurons and cardiomyocytes. Given the abundance of cardiac nerves and their critical role in arrhythmia formation (Ripplinger et al. 2016, Zhu et al. 2022), further investigation into their potential ephaptic interactions is warranted.

Referee #2:

The submitted review article by Drs Wu, Payne, and Gourdie provides a comprehensive review on homo- and heterocellular electrical coupling and the proposed underlying mechanisms, summarizing the relevant literature and the current state-of-knowledge.

Please find my detailed suggestions for improving the manuscript (minor revisions) below.

- The title of the manuscript states "electrical interactions", whereas to my understanding the provided manuscript focusses almost exclusively on mechanisms of gap-junction-mediated and ephaptic electrical coupling. I would therefore suggest to further focusing the title of the manuscript. (For example, aspects such as electrical insulation by fibroblasts or neuron-supporting cells are not discussed, also the functional relevance of electrical interactions for the individual cell types beyond their influence on cardiac impulse propagation is not discussed in detail.)

We thank the reviewer for their thoughtful feedback on the title. We agree that the original title was too broad and did not accurately reflect the focus of our review, which is on gap junctional and ephaptic coupling mechanisms. To address your concern, we have revised the title to more accurately convey the scope. The revised title is:

Gap Junctional and Ephaptic Coupling in Cardiac Electrical Propagation: Homocellular and Heterocellular Perspectives

We hope this addresses your concerns.

- The manuscript discusses in detail a recent study showing Cx-dependent and independent coupling between cardiac scar-resident fibroblasts and cardiomyocytes (Wang et al., reference 12). While the data presented in this study is indeed intriguing, the study has been very controversially discussed, for example because of light intensities used that are 100-1000-fold higher than those necessary to effectively stimulate Channelrhodopsin-2, but also because the computational modelling data builds on parameters (e.g. Na channel conductivity in fibroblasts), that are beyond values assumed to be realistic in the field. As already highlighted in the current manuscript, the presence of perinexus-like structures at the interfaces between fibroblasts and cardiomyocytes has not been shown, thus the mechanisms underlying the Wang study are unclear at present. I would therefore suggest to even more cautiously discuss data from this paper, highlighting limitations of the study and lack of mechanistic understanding.

Thank you for the feedback. We agree that the Wang et. al. study, while intriguing, leaves several key questions unaddressed. The concerns raised have deepened our understanding of this contribution and the need for further work. Regarding light intensity, we agree that it appears to be higher than in other reports (PMID: 30618818; PMID: 35488105). Whilst this intensity is notable, it was not extreme relative to previous studies, so we'd rather emphasize the need for further understanding of unknowns on the molecular mechanism and the need for further refinement of their computational model. We view that a synthesis of published data, together with pointing to broader, big picture implications of published work, as well as gaps in knowledge, is appropriate for a literature review. Specific critiques of methodological approach at the granular level, in our view, is better tackled as part of peer-reviewed publications with experiments so that opinions are backed by data. We have nonetheless revised the relevant section of the manuscript in response to the reviewer as follows:

A computational model was used to further investigate electrical interactions between cardiomyocytes and fibroblasts, which indicated that GJC and EpC may excite myocytes coupled to fibroblasts in a synergistic, yet functionally redundant manner.

This study provides new insights into the mechanisms of GJC and EpC between cardiomyocytes and fibroblasts in vivo. However, several questions remain including: 1) What are the specific nanostructures responsible for this form of heterocellular EpC? ; 2) Can scar fibroblasts form shared VGSCs complexes with cardiomyocytes? ; and 3) Do narrow perinexus-like clefts of extracellular space form between fibroblasts and cardiomyocytes? Also, whilst the current computational model provides an initial investigative framework, its architecture will likely need to undergo iterative refinement as mechanistic insights into fibroblast-myocyte EpC dynamics evolve. Key parameters—notably sodium channel conductivity in fibroblast populations—warrant

targeted validation through both sensitivity analyses and empirical electrophysiological studies to further strengthen predictive accuracy.

- The in-depth discussion of the proposed mechanisms of ephaptic coupling is important. However, in the current form, I cannot follow the discussion on "Na ion depletion" in the perinexus. Has this been experimentally validated, e.g. by use of Na-concentration dependent fluorophores or Na-selective electrodes? Given the very high driving force for Na import, it is hard to imagine that the gradient can be depleted (only very few ions need to be transported for large transmembrane voltage changes to occur), even within diffusion-limited nanodomains. Would one not also expect a change in extracellular K concentration? Another interesting aspect in this regard is how the perinexus is organised in species with different heart rates (e.g. mouse vs human)?

We sincerely thank you for the insightful thoughts regarding the mechanisms of ephaptic coupling. Current investigations into ephaptic coupling are primarily limited to computational modelling and whole-heart studies. Unfortunately, the dynamics of sodium ions and changes in cleft potential within the perinexus have not yet been experimentally validated. Furthermore, computational modeling suggests that voltage-gated potassium channel currents in the perinexus also undergo changes in response to ephaptic coupling, a point we have discussed in this manuscript. Therefore, we have little experimental validation of ephaptic coupling mechanisms at cellular level. Hopefully, with the advancement of techniques, we will be able to probe ion dynamics in the perinexus. Regarding perinexal remodeling in humans, the available data is currently limited to a single report demonstrating that the atrial perinexus widens with aging and in patients with atrial fibrillation relative to healthy individuals (PMID: 29780324). Consistently, studies across several vertebrate species, including mouse (PMID: 38214908), guinea pig (PMID: 30106376, PMID: 34623182 and PMID: 40047771), and humans (PMID: 29780324), indicate that the perinexus maintains a similar width range of 20 and 30nm. We appreciate the reviewer's attention to this topic and hope this clarification addresses your concerns.

- The manuscript would benefit from a short discussion on methodological advances that would facilitate future investigations into electrical coupling mechanisms. Which methods need further development or could be adapted already to fill open knowledge gaps? Some suggestions: Nanodomain functional imaging of diffusion or changes in ion concentrations, 3D structural methods at nano- and microscale to characterize cell-cell interaction sites, optogenetic actuators targeted to specific cellular compartments etc.

Thank you for your valuable suggestion. We included a paragraph in the manuscript to discuss future methodologies. This paragraph is provided below here for your convenience. We hope this addresses your concerns.

This being said, current techniques impose significant limitations on our ability to fully understand the mechanisms of electrical coupling, particularly ephaptic coupling, at the cellular level. One major challenge is determining whether gap junctional and ephaptic

couplings function as truly independent mechanisms. Existing techniques cannot ensure the complete knockout of all connexin types involved in gap junctional coupling within cardiac tissue. Although CRISPR-Cas9 gene-editing is a promising tool, achieving the knockout of all connexins in cardiomyocytes or other relevant cell types remains challenging. Moreover, there is currently no method available to directly track sodium ion diffusion into the perinexus. A potential approach could involve a modified Fluorescence Resonance Energy Transfer (FRET) system, where labeled sodium ions act as donors and perinexal sodium channels serve as acceptors, allowing visualization of sodium ion movement near these channels. Further advancements in imaging, electrophysiology, and molecular tools are essential for overcoming these limitations and gaining deeper insights into ephaptic coupling mechanisms.

- In certain places, the manuscript would benefit from more clearly defined terminology and/or by the specification of quantitative measures.

Thank you for your suggestion. Please see my responses to each of your points below.

- o Line 25: Please specify "heterocellular structures"

We revised the sentence to specify that heterocellular structures indicate different cell types:

gap junctions are also present between different cell types in heterocellular structures

- o Line 44: You may want to reconsider the term "communication", I would suggest something like "electrical impulse propagation".

Replaced.

- o Line 47: "Electrical propagation" - please specify what is propagated, e.g. "action potentials" or "depolarization/repolarization cycles".

Changed.

- o Lines 82-84: Please provide quantitative values for low and high conductances

The conductance for Cx40, Cx43 and Cx45 with references is provided in the manuscript. The Cx40, Cx43 and Cx45 - formed gap junction channels have conductance of ~180pS, 90-120pS and 30-40pS, respectively.

- o Line 93: You may want to add a statement of intercalated disc orientation and directionality of conduction here. You later introduce lateralization of connexins, but this

may not be clear without introducing the normal distribution of connexins in myocytes (e.g. ratio between short and long axes).

A revision has been made to state the intercalated disc orientation and directionality of conduction and it is provided here for your convenience:

GJs are primarily located within the interplacate of mature intercalated disc, which is oriented perpendicular to the long axis of the cell, particularly at the disc periphery, facilitating longitudinal conduction of electrical impulses (Gourdie et al. 1991).

o Line 107: Can you please specify "reductions in CV"? Are these due to changes in connexin expression, localization, and/or conductivity of individual gap junctions?

The reductions in CV is caused by decreased gap junction conductance. To clarify, this sentence has been revised in the manuscript and is provided here:

Mutations in the gene encoding Cx43, GJA1, can reduce Cx43 phosphorylation and induce loss of function of GJs, leading to decreased GJ conductance. This, in turn, slows conduction and increases the risk of cardiac arrhythmogenesis (Paznekas et al. 2003, Kalcheva et al. 2007, Van Norstrand et al. 2012).

o Lines 126/127: "reports of both structural GJs and conductance and electrical activity"
- Please phrase more clearly.

This sentence has been revised in the manuscript and it is provided here:

Many studies of fibroblast-fibroblast electrical interactions have been undertaken in cell culture, with demonstrations of gap junctions between fibroblasts by electron microscopy and measurements of channel conductance levels at ~22 pS (Rook et al. 1989, Doble et al. 1995, Kohl et al. 2014) —significantly lower than the 90-120 pS conductance found for gap junctions formed by Cx43 alone (Elenes et al. 2001).

o Lines 137: Please specify comparison: "showed increased GJC" compared to XXX.

The sentence is revised to show the comparison. Here it is:

Intercellular communication between fibroblasts isolated from infarcted and non-infarcted regions, assessed by Lucifer yellow dye, was increased compared to fibroblasts isolated from sham-operated hearts.

o Lines 145: "techniques and approach" - Please correct grammar and please specify the techniques you are referring to.

This sentence has been revised according to your suggestion. Here it is:

With the development of new visualization techniques that improve resolution of the rare and small gap junctions that form between fibroblasts in native tissue, we may be able to better address such questions, improving our understanding of GJ structure and function between cardiac fibroblasts.

o Lines 156/157: "Fibroblasts are expressed in cultured cardiomyocytes" - Please rephrase (co-expression model).

It is rephrased as: *fibroblasts are co-cultured with cardiomyocytes.*

o Line 164: "Active participants" - term could be misleading. Do you refer to the presence of voltage-gated Na channels in fibroblasts? Otherwise I would use different terminology to highlight the role of fibroblasts for cardiac impulse propagation (not initiation).

This sentence is revised as:

...likely play a functional role in the heart's electrical network through their GJ connections with cardiomyocytes

o Lines 175: Would replace "scar" by "scar-resident".

Replaced.

o Line 179: I would propose to properly introduce Channelrhodopsin-2 here (currently only introduced on line 200) - light-gated cation non-selective channel (Channelrhodopsin-2)

It has been revised. Here it is:

... a mouse that expressed a light-gated cation non-selective channel (Channelrhodopsin-2, ChR2) ...

o Line 185: "is" - would suggest to replace by "can be".

Replaced.

o Line 221: "Cx43 expression significantly increased" - please specify in which cell type.

In this study, Cx43 expression was evaluated by immunofluorescence. They found that Cx43 expression between cardiomyocytes and endothelial cells increases compared to cardiomyocytes alone. So, we revised this sentence:

Cx43 expression between cardiomyocytes and endothelial cells significantly increased compared to cardiomyocytes cultured alone...

o Lines 223/224: "suggesting a role for GJs between these two cell types in spread of electrical activation" - description of evidence and thus conclusion is not very clear. Is there any direct evidence of (gap junctional) coupling between the two cell types, or could the interaction be indirectly (e.g. signalling based)?

Thank you for the insightful thoughts. The existence of gap junctional coupling between cardiomyocytes and endothelial cells remains unknown. It is the only study that investigates Cx43 between these two types of cells (Narmoneva *et al.* 2004). To address your concerns, we revised the text:

suggesting the formation of GJs between these two cell types. However, it remains unclear if gap junctional coupling occurs between these two types of cells in native tissue.

o Lines 229/230: "primary mechanism for electrical interaction between excitable cells" - see comment on title; I think interaction is too general and then the statement might be debatable.

The 'interaction' is replaced with 'impulse propagation'

o Line 235: "sufficiently close proximity" - please specify.

"sufficiently close proximity" is replaced with 'within nanometer-scale proximity'

o Line 279: "mildly widened" - please specify. In the schematic Figure, there are two charges more than for normal, but what this represents in terms of distance/area/volume or actual numbers of charges is not very clear.

Kucera *et al* established the first computational model regarding conduction velocity vs cleft width (equivalent to perinexal width) in 2002. This model predicts that conduction velocity has a biphasic response to the width of the cleft. The 'mildly widened' refers to the perinexus with width less than 50nm. We have revised the text to reflect the width of mildly widened perinexus and it is provided here:

According to the 2002 computational model of Kucera and colleagues, when the perinexus is mildly widened from 20 nm (normal) to 50 nm, and gap junctional coupling is at the normal level representing 3-5%, the increased amount of sodium ions due to perinexal expansion can lead to an increase in sodium current in both cells, as well as

the decrease in cleft potential V_p , which enhances cardiac conduction (Kucera et al. 2002).

Furthermore, thank you for pointing out the inappropriate number of charges under the mildly widened condition compared to normal condition in the schematic Figure. We have revised the Figure to reflect the increased number of charges under mildly widened condition. These increased number of charges basically represent the consequence caused by the perinexus expansion as the expanded perinexal volume allows increased number of sodium ions compared to the normal perinexus condition. We hope our explanation addresses your concerns.

o Line 312: Would the Na channel density in fibroblasts be high enough for effective ephaptic coupling?

Theoretically, Na channel density must be high enough for ephaptic coupling according to the computational model of cardiomyocytes (PMID: 12480819). Since EpC between fibroblasts has not yet been investigated, it remains unknown if Na channel density between fibroblasts is sufficient for effective ephaptic coupling.

o Line 330: Please use proper terminology when discussing findings on ChR2 (see comment above).

Changed.

o Line 337: "these workers" - please rephrase

This sentence is revised. It starts with: *A computational model was established to...*

o Line 341/342: Please reconsider phrasing of statement as study is highly controversial (see comment above).

We revised this statement to highlight the following key concerns and questions unaddressed in this study. Here it is:

This study provides new insights into the mechanisms of GJC and EpC between cardiomyocytes and fibroblasts in vivo.

- The Figures could be improved to further support the presented concepts.

o Figure 1B: Please double-check scale bars of overview images and insets. The inset on the left seems to be more than 2-fold increased compared to the overview image. Does the highlighted region correspond 1:1 to the zoom-in?

We agree that the zoomed-in image appears to be 2-fold larger than the highlighted region. We have placed the correct 250 nm scale on the image.

o Line 404: "is narrow" - Please correct grammar and specify dimension if possible.

Corrected by adding 'a' before 'narrow'

o In the context of Figure 1 you may want to highlight the need for 3D nanoscopic investigations to better understand diffusion-limited domains, their access and volumes.

We included a statement in the manuscript according to your suggestion. Here it is:

Importantly, a detailed examination of ion dynamics in the perinexus within its 3D structure will enhance our understanding of the diffusion process, such as how sodium ions access the perinexus during self-attenuation, thereby elucidating the mechanisms of ephaptic coupling at the nanoscale.

o Figure 2: The difference between the upper and middle row is difficult to grasp, and the faster conduction (middle row) is not shown directly. Providing a voltage value always requires a reference, you may want to add this to the legend.

Thank you for pointing this out. We have revised the figure and its legend. In the updated figure, there should be no available sodium ions flowing into Cell 2 when the perinexal width is in the normal range due to depletion of the perinexal sodium ions. We also included text to indicate the corresponding changes in the conduction at different perinexal width. In addition, we included a brief explanation of sodium ions dynamics and related changes when the perinexus is widened. We hope these changes address your concerns and provide the readers with a clear understanding of the mechanisms underlying ephaptic coupling.

o Figure 3: Please use more realistic representations of cardiac interstitial cells, as well as endothelial cells and axons (not necessarily terminating on cardiomyocytes, but having axonal contacts along their entire length). In intact myocardium, these cells all form relatively thin processes wrapping around cardiomyocytes (filling the "gaps"). E.g. see <https://pubmed.ncbi.nlm.nih.gov/37674983/>
<https://www.biorxiv.org/content/10.1101/2023.11.30.569388v1>

Thank you for your suggestion. We have revised Figure 3 to better represent different types of cells. The references you referred to is very helpful for making such revision. Please let me know if the additional revision is needed.

• Finally, there are a numbers of typos to be corrected.

o Line 91: please remove extra "and"

Removed

o Line 126: please remove extra comma

Removed

o Line 216: "play role"

'a' is added before 'role'

o Line 219: "That formation" (incomplete sentence)

'That' is replaced with 'The'

o Line 249: Please correct: "Kucera"

Corrected.

o Line 260: Extra space.

Removed.

o Figure 1A: Please add space between number and unit.

Corrected

REQUIRED ITEMS

- Please include an Abstract Figure file, as well as the Figure Legend text within the main article file. The Abstract Figure is a piece of artwork designed to give readers an immediate understanding of the Review Article and should summarise the main conclusions. If possible, the image should be easily 'readable' from left to right or top to bottom. It should show the physiological relevance of the Review so readers can assess the importance and content of the article. Abstract Figures should not merely recapitulate other figures in the Review. Please try to keep the diagram as simple as possible and without superfluous information that may distract from the main conclusion of the Review. Abstract Figures must be provided by authors no later than the revised manuscript stage and should be uploaded as a separate file during online submission labelled as File Type 'Abstract Figure'. Please ensure that you include the figure legend

in the main article file. All Abstract Figures will be sent to a professional illustrator for redrawing and you may be asked to approve the redrawn figure before your paper is accepted.

Since Figure 3 summarizes the content of this review article, it also serves well as the Abstract Figure. The Abstract Figure has been submitted.

The figure legend for Abstract Figure can be found between Figure 3 and References. It is also provided here:

Abstract Figure: Intricate cellular electrical coupling networks in the heart. Various cell types couple the central cardiomyocyte through gap junctional contacts, with the exception of neurons. Whether ephaptic coupling (EpC) occurs in homocellular or heterocellular contexts beyond cardiomyocyte-cardiomyocyte interactions remains unclear.

- Your MS must include a complete "Additional information section" with the following 4 headings and content:

Competing Interests: A statement regarding competing interests. If there are no competing interests, a statement to this effect must be included. All authors should disclose any conflict of interest in accordance with journal policy.

The section of competing interests is added.

Author contributions: Each author should take responsibility for a particular section of the study and have contributed to writing the paper. Acquisition of funding, administrative support or the collection of data alone does not justify authorship; these contributions to the study should be listed in the Acknowledgements. Additional information such as 'X and Y have contributed equally to this work' may be added as a footnote on the title page.

It must be stated that all authors approved the final version of the manuscript and that all persons designated as authors qualify for authorship, and all those who qualify for authorship are listed.

The section of author contributions is included.

Funding: Authors must indicate all sources of funding, including grant numbers. If authors have not received funding, this must be stated.

The section of Funding is added.

It is the responsibility of authors funded by RCUK to adhere to their policy regarding funding sources and underlying research material. The policy requires funding information to be included within the acknowledgement section of a paper. Guidance on how to acknowledge funding information is provided by the Research Information Network. The policy also requires all research papers, if applicable, to include a statement on how any underlying research materials, such as data, samples or models, can be accessed. However, the policy does not require that the data must be made open. If there are considered to be good or compelling reasons to protect access to the data, for example commercial confidentiality or legitimate sensitivities around data derived from potentially identifiable human participants, these should be included in the statement.

Acknowledgements: Acknowledgements should be the minimum consistent with courtesy. The wording of acknowledgements of scientific assistance or advice must have been seen and approved by the persons concerned. This section should not include details of funding.

The section of Acknowledgements is added.

- The reference list must be in alphabetical order, rather than numbered, to comply with our Journal format.

The reference has been revised in order to comply with the journal format.

- Please upload separate high quality figure files via the submission form.

All figures have been submitted.

- Author profile(s) must be uploaded via the submission form. Authors should submit a short biography (no more than 100 words for one author or 150 words in total for two authors) and a portrait photograph of the two leading authors on the paper. These should be uploaded and clearly labelled together in a Word document with the revised version of the manuscript. Any standard image format for the photograph is acceptable, but the resolution should be at least 300 DPI and preferably more. A group photograph of all authors is also acceptable, providing the biography for the whole group does not exceed 150 words.

A word document including the author profiles has been submitted.

END OF COMMENTS

Dear Dr Gourdie,

Re: JP-TR-2025-287358R1 "Gap Junctional and Ephaptic Coupling in Cardiac Electrical Propagation: Homocellular and Heterocellular Perspectives" by Xiaobo Wu, Laura Beth Payne, and Robert G. Gourdie

We are pleased to tell you that your paper has been accepted for publication in The Journal of Physiology.

Authors should note that it is too late at this point to offer corrections prior to proofing. Major corrections at proof stage, such as changes to figures, will be referred to the Editors for approval before they can be incorporated. Only minor changes, such as to style and consistency, should be made at proof stage. Changes that need to be made after proof stage will usually require a formal correction notice.

Yours sincerely,

Bjorn Knollmann
Senior Editor
The Journal of Physiology

P.S. - You can help your research get the attention it deserves! Check out Wiley's free Promotion Guide for best-practice recommendations for promoting your work at www.wileyauthors.com/eoo/guide. You can learn more about Wiley Editing Services which offers professional video, design, and writing services to create shareable video abstracts, infographics, conference posters, lay summaries, and research news stories for your research at www.wileyauthors.com/eoo/promotion.

IMPORTANT NOTICE ABOUT OPEN ACCESS: To assist authors whose funding agencies mandate public access to published research findings sooner than 12 months after publication, The Journal of Physiology allows authors to pay an Open Access (OA) fee to have their papers made freely available immediately on publication.

You can check if your funder or institution has a Wiley Open Access Account here: <https://authorservices.wiley.com/author-resources/Journal-Authors/licensing-and-open-access/open-access/author-compliance-tool.html>.

EDITOR COMMENTS

Reviewing Editor:

The authors have addressed all of the reviewers' concerns.

Senior Editor:

The MS is acceptable. Thank you for your excellent contribution to the Journal!

REFEREE COMMENTS

Referee #1:

The authors have addressed all of my concerns. No additional comments.

Referee #2:

Thank you very much for providing the revised manuscript. All reviewer comments have been adequately addressed. The overview article will serve as a useful resource for the community, providing a timely, critical, and balanced review of the mechanisms underlying cell-cell coupling in the heart.